# Cell-free tumor DNA, CA125 and HE4 for the objective assessment of tumor burden in patients with advanced high-grade serous ovarian cancer

Florian Heitz[1,2,3,4]*, Sotirios Lakis[5,6], Philipp Harter[1], Sebastian Heikaus[7], Jalid Sehouli[2,3,4], Jatin Talwar[5], Roopika Menon[5], Beyhan Ataseven[1,8], Miriam Bertrand[5,9], Stephanie Schneider[1], Erika Mariotti[5], Mareike Bommert[1], Judith N. Müller[5], Sonia Prader[1,10,11], Frauke Leenders[5], Alexandra Hengsbach[1], Christian Gloeckner[5], Elena Ioana Braicu[7], Lukas C. Heukamp[5], Andreas du Bois[1], Johannes M. Heuckmann[5,12]

**1** Evangelische Kliniken Essen-Mitte Klinik für Gynäkologie und gynäkologische Onkologie, Essen, Germany, **2** Department for Gynecology with the Center for Oncologic Surgery Charité Campus Virchow-Klinikum, Charité–Universitätsmedizin Berlin, Berlin, Germany, **3** Corporate member of Freie Universität Berlin, Berlin, Germany, **4** Berlin Institute of Health, Humboldt-Universität zu Berlin, Berlin, Germany, **5** NEO New Oncology GmbH, Cologne, Germany, **6** ULTIVUE, Segrate Milan, Italy, **7** Evangelische Kliniken Essen-Mitte, Center for Pathology, Essen, Germany, **8** Department of Obstetrics and Gynecology, University Hospital, LMU Munich, Munich, Germany, **9** Institute of Medical Genetics and Applied Genomics, University of Tübingen, Tübingen, Germany, **10** Department of Obstetrics and Gynecology, General Hospital (SABES-ASDAA), Brixen–Bres-sanone, Italy, **11** Department of Obstetrics and Gynecology, Innsbruck Medical University, Inns-bruck, Austria, **12** PearlRiver Bio GmbH, Dortmund, Germany

* florian.heitz@gmx.net

## Abstract

### Background

The present prospective study aimed at determining the impact of cell-free tumor DNA (ct-DNA), CA125 and HE4 from blood and ascites for quantification of tumor burden in patients with advanced high-grade serous epithelial ovarian cancer (EOC).

### Methods

Genomic DNA was extracted from tumor FFPE and ct-DNA from plasma before surgery and on subsequent post-surgical days. Extracted DNA was subjected to hybrid-capture based next generation sequencing. Blood and ascites were sampled before surgery and on subsequent post-surgical days. 20 patients (10 undergoing complete resection (TR0), 10 undergoing incomplete resection (TR>0)) were included.

### Results

The minor allele frequency (MAF) of *TP53* mutations in ct-DNA of all patients with TR0 decreased significantly, compared to only one patient with TR>0. It was not possible to distinguish between patients with TR0 and patients with TR>0, using CA125 and HE4 from blood and ascites.

**Data Availability Statement:** All relevant data are within the manuscript and its Supporting Information files.

**Funding:** This study was supported by NewOncology, Cologne, Germany. The funders played a role in study design, and analysis, and preparation of the manuscript.

**Competing interests:** FH: Travel grants: AstraZeneca, Tesaro, Roche; Honoraria: Roche, AstraZeneca; Clovis, Advisory: Roche; SL: personal fees from NEO New Oncology GmbH, personal fees from BioNTech Diagnostics, personal fees from Definiens GmbH; PH: Honoraria: Roche, AstraZeneca, Tesaro; Advisory: Roche, AstraZeneca, Tesaro, PharmaMar, Lilly; SH: none; JS: HONORARIA: Astra Zeneca, Eisai, Clovis, Olympus, Johnsons and Johnson, PharmaMar, Pfizer, TEVA, TESARO, MSD; CONSULTING OR ADVISORY ROLE: Astra Zeneca, Clovis, Lilly, PharmaMar, Pfizer, Roche, TESARO, MSD; RESEARCH FUNDING: Astra Zeneca, Clovis, Merck, Bayer, PharmaMar, Pfizer, TESARO, MSD; TRAVEL, ACCOMODATIONS, EXPENSES: Astra Zeneca, Clovis, PharmaMar, Roche, Pfizer, TESARO, MSD; JT: employed at New Oncology; RM: employed at New Oncology; BA: reports receiving honoraria from Roche, Tesaro, Clovis, AstraZeneca, and Celgene for lectures, and is an unpaid consultant/advisory board member for Roche and Amgen; MB employed at New Oncology; SS: none; EM: employed at New Oncology; MB: Travel support from prIME Oncology; JNM: employed at New Oncology; SP: none; FL: employed at New Oncology; AH: none; CG: employed at New Oncology; EIB: reports receiving honoraria for advisory board and educational activities from AstraZeneca, Clovis, Tesaro, GSK, Roche Pharma, Incyte, Eisai, MSD, Abbvie; reports receiving travel costs from Clovis, Tesaro, Roche Pharma; LCH: employed at New Oncology; AdB: reports honorary for advisory board and educational activities for Roche, Astra Zeneca, Tesaro, Clovis, Biocad, and Genmab; JMH: employed at New Oncology. That the competing interests does not alter our adherence to PLOS ONE policies on sharing data and materials.

## Conclusions

Based upon the present findings, ct-DNA assessment in patients with high-grade serous EOC might help to better determine disease burden compared to standard tumor markers. Further studies should prospectively evaluate whether this enhancement of accuracy can help to optimize management of patients with EOC.

## Introduction

Epithelial ovarian cancer (EOC) is the second most frequent malignancy of the female genital system and the most fatal gynecologic cancer in developed countries [1]. Approximately two thirds of women with newly diagnosed EOC present with advanced disease [2,3]. Debulking surgery and chemotherapy are the cornerstones of treatment for patients with EOC. After optimal treatment for primary EOC, e.g. complete resection, platinum-based chemotherapy, and maintenance therapy with olaparib in patients with pathologic *BRCA* mutations, 25% of patients will experience recurrent disease within 2 years and will subsequently die. In patients with residual disease left after surgery, and/or suboptimal systemic therapy ~75% of patients will experience recurrent disease within 2 years [4]. Therefore, further improvement of therapy, but also of evaluation of response to treatment, to individually tailor therapy, is essential. Performance status and symptoms, split-image procedures, and evaluation of tumor markers are being used to monitor and tailor systemic therapy of patients with EOC. CA125 (MUC-16) is the best-evaluated tumor marker in EOC. The transmembrane glycoprotein is regularly elevated in serum of patients with EOC and it supports diagnosis and guides treatment of patients with EOC [5,6]. Human epididymis protein 4 (HE4) is overexpressed in EOC cells and several studies reported good performances of circulating HE4 for EOC detection [7,8] and recent reviews highlighted its role as a prognostic biomarker [9,10]. Cell-free tumor DNA (ct-DNA) analyses may overrule performance of established biomarkers as several genes are being evaluated and sensitivity might also be improved [11,12]. EOC is composed of different histological subtypes which harbor distinct mutational landscapes [13] and high-grade serous histology is the most frequent subtype [14]. High-grade serous EOC is molecularly characterized by the presence of *TP53* mutations in tumor DNA [15–17] and *TP53* mutations can be found in ct-DNA of patients with high-grade serous EOC [18]. Moreover, mutations in genes detected in ct-DNA have been associated with disease burden at the onset of chemotherapy in patients with high-grade EOC and an early decline of *TP53* mutant allele frequency in ct-DNA after one cycle of chemotherapy was correlated with time to progression [18]. This prospective proof-of-principle study was designed to assess ct-DNA as quantitative measure of tumor burden in patients with high-grade serous EOC and to compare it to the established markers CA125 and HE4.

## Methods

### Patients and sample collection

All subjects gave their written informed consent for inclusion before participation in the study. The study was conducted in accordance with the Declaration of Helsinki, and the protocol was approved by the Ethics Committee of the Landesärztekammer Nordrhein (No.: 2015377). Patients with treatment-naïve known or highly suspected advanced (FIGO IIIC or IV) high-grade serous EOC were scheduled for primary debulking surgery in the Department

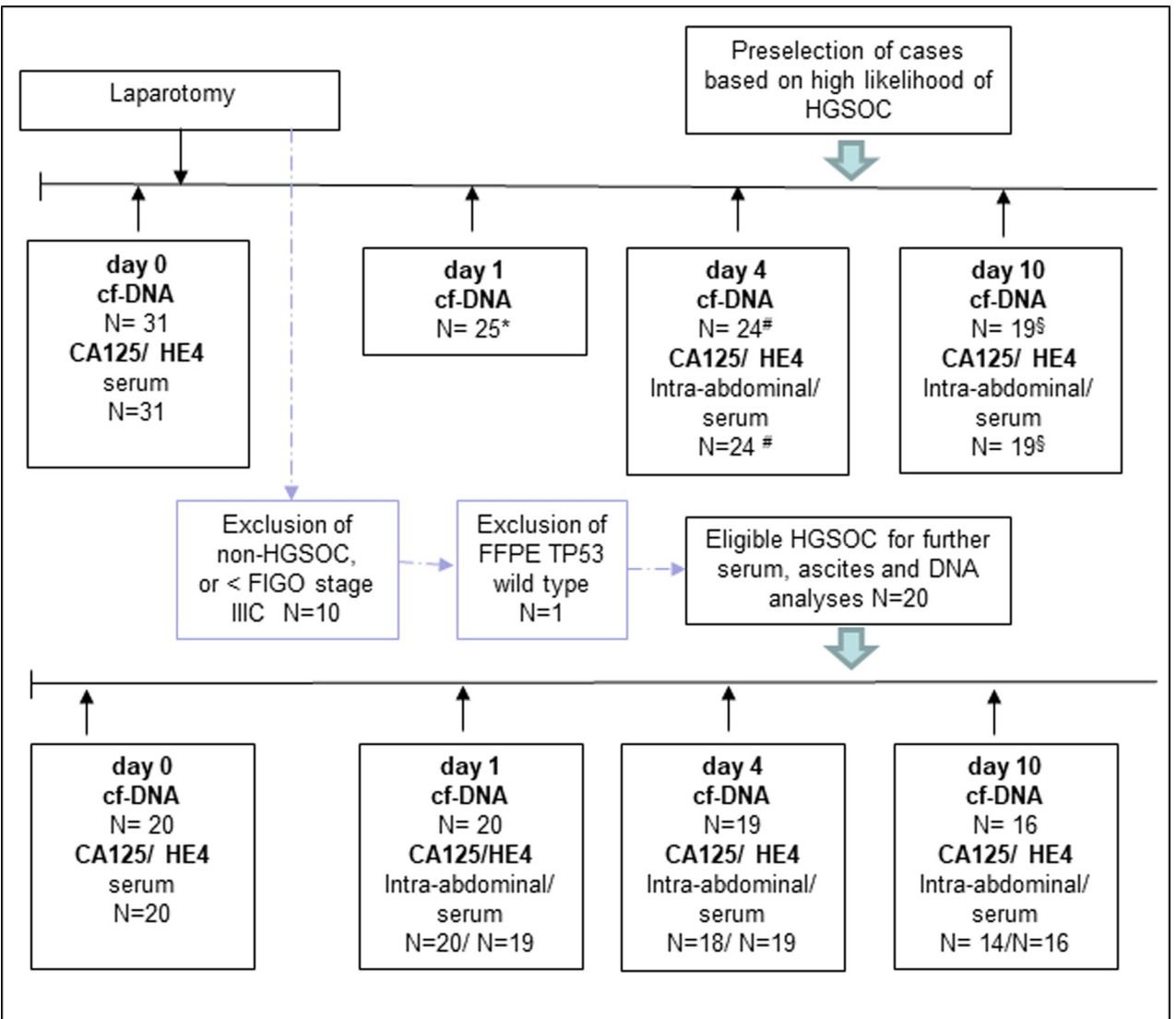

**Fig 1. In-/Exclusion of patients and sample availability.** Sample available at each time point, finally 10 patients with complete resection, and 10 patients with residual disease; CTX: Chemotherapy, *6 patients had to be excluded due to the following reasons at several time points: *1 endometrioid OC, 5 benign disease; # 1 patient withdrawl consent for further sampling at d4; § 5 patients had been discharged from hospital already.

for Gynecology and Gynecologic Oncology of the Evangelische Kliniken Essen-Mitte, Germany. A detailed description of the general surgical approach of our department is reported elsewhere [19]. A non-high-grade serous pathological diagnosis resulted in a drop-out.

Peripheral venous blood was collected in Cell-Free DNA BCT tubes on days 0 (before surgery) and on post-operative days 1, 4, and 10. Samples were immediately shipped at room temperature to NEO New Oncology GmbH (Cologne, Germany) for preparation of plasma and DNA extraction. DNA was stored until confirmation of diagnosis and discarded for all non-high-grade serous EOCs. Additional blood was drawn preoperatively and at days 4, and 10 after surgery for assessing CA125 and HE4. Intraperitoneal concentrations of CA125 and HE4 were determined in samples of ascites from intra-abdominal drainage intraoperatively, and on post-operative days 1, 4 and 10. Tumor tissue was collected at debulking surgery and was fixed in formalin- and embedded in paraffin. Fig 1 displays the timeline of sample collection.

Reasons for incomplete resection were: miliary disease on the visceral serosa of the small-bowl, infiltrating tumor on the mesenteric root, or tumor at the hepatic-duodenal ligament. For the purpose of this proof-of-principle study the determination of "tumor burden" was based upon the post-surgical result of debulking surgery. Patients without macroscopic residual disease were defined as TR0 ("low tumor burden"), patients with any macroscopic residual disease were defined as TR>0 ("high tumor burden"). An analogue definition has been published earlier [20]. No cancer-specific systemic therapy was given to the patients during surgery or in the post-surgical days during the study.

**Sample preparation.** Pathological diagnosis of high-grade serous EOC was confirmed by conventional pathology and immunohistochemistry. The antibody reactions were done by the UltraView DAB Kit (Ventana). The immune-reactions of the antibodies p53 (DO-7), WT-1 (6F-H2), p-16 (E6H4), CK7 (SP52) and PAX-8 (MRQ-50) (all purchased from Roche Diagnostics, Swiss) were analyzed semi quantitatively. After pathological review, genomic DNA was extracted from 5 to a maximum of 15 sections (10 μm) of FFPE material, with a tumor content of 10% or more. Genomic DNA was extracted from FFPE material,and sheared (Covaris) using a semi-automated extraction protocol (Maxwell®16, Promega). Whole blood (18ml) was collected in Streck tubes (Cell-Free DNA BCT, Streck, Ref. 218997) and cell free DNA (cf-DNA) was extracted using Qiagen's QIAamp Circulating Nucleic Acid kit (QIAGEN, Cat.-No. 19419). For the purpose of DNA extraction, plasma from 3-15ml was obtained depending on the sample. Fragmented DNA was subjected to hybrid-capture based next-generation sequencing to detect point mutations, small insertions and deletions, copy number alterations and genomic translocations (both NEO New Oncology GmbH, Cologne, Germany). In brief, after cf-DNA extraction, adapters unique molecular identifiers were ligated and individual genomic regions of interest were enriched using complementary bait sequences (hybrid-capture procedure). The selected baits ensure optimal coverage of all relevant genomic regions. Post enrichment, targeted fragments were amplified (clonal amplification) and sequenced in parallel aiming for a sequencing depth averaging 2500x. After extraction the fragment size for each sample was determined by a distinct peak which was obtained by running the samples through a Fragment Analyser (Agilent). The instrument uses a capillary electrophoresis-based separation technique and provides peaks based on the fragment sizes present in the sample. The fragment size of the cfDNA ranged between 160-180bp based on the nucleosome cleavage sites. The amount of cfDNA in the sample includes both DNA that is shed from the tumor (ctDNA) and normal cells. A clear indication of the tumor DNA in the sample can be judged by the MAFs. However, recent publications have also shown the utility of simple cfDNA concentration measurement [20], thus mutation allele frequency is a more precise measure of residual tumor compared to cfDNA and offers the advantage of genotyping. Computational analysis was performed using NEO New Oncology's proprietary computational biology analysis pipeline to remove sequencing artifacts and detect relevant genomic alterations in a quantitative manner. As the purpose of the study was primarily to examine the presence and the dynamics of cancer genotypes in the circulation, we only assessed genomic alterations in the tissue DNA and cf-DNA that could be monitored by the gene panel established for cell-free tumor DNA (ct-DNA) analyses (S1A and S1B Table). Somatic mutations, including small indels were identified with a stepwise approach. Variants were excluded as SNPs if present in the dbSNP, ESP6500 and 1K genome data bases. Additional variants were filtered out as germline by comparing MAF (minor allele frequency) in FFPE and the corresponding plasma samples. The NEOliquid test was designed to identify genes either with a direct or indirect impact on patient treatment decisions for solid tumors. Therefore, core bioinformatics analysis was set up to exclude variants with MAF in the range of 50% or 100% which are highly unlikely to be somatic in origin. Manual curation was needed only in select cases. In all cases, MAF of such

variants differed markedly from cancer-related mutations. The panel of genes for the NEOliquid test were part of a CE kit produced by NEO New Oncology GmbH.

*Statistics.* The unpaired two-sided wilcoxon rank sum test was used to determine the significance of changes across multiple days for patients paired by debulking status. A non-parametric test was choosen as the data was skewed towards low numbers (because of the decrease in MAF and serum and ascites values). To analyse the CA125 and HE4 data, values were logarithmized due to non-normal distribution. The Pearson correlation coefficient was used to measure the correlation between two sets of data. Analyses were conducted using the rstatix R-package.

## Results

### Patients' characteristics

Thirty-one patients were recruited. Fig 1 displays the reason for, and numbers of patients excluded. Ten patients with macroscopic complete resection (TR0) and ten patients with residual disease (TR>0) -fulfilling the inclusion criteria- were included. The clinicopathologic characteristics are summarized in Table 1.

**Table 1. Patients' characteristics.**

| Pat No. | TR | FIGO | pathological details | Ascites (ml) |
|---|---|---|---|---|
| KEM-002 | 0 | IVB | pT3c, pN1 (3/74), pL1, pV0, pM1b (CPLN) | 500 |
| KEM-008 | 0 | IVB | pT3c, pN1(17/68), pL1, pV0, pM1b (CPLN) | 1800 |
| KEM-013 | 0 | IVB | pT3c, pN0 (0/23), pL1, pV0, pM1b (LSK) | 600 |
| KEM-015 | 0 | IIIC | pT3c, pN0 (0/83), pL0, pV0 | 100 |
| KEM-016* | 0 | IVB | pT3c, pN1b (13/92) pL1, pV0, pM1b (LSK, CPLN) Uterine carcinoma pT1a, G1 | 600 |
| KEM-021 | 0 | IIIC | pT3c, pN1 (4/78), pL1, pV0 | 6000 |
| KEM-023 | 0 | IVB | pT3c, pN1 (4/99), pL1, pV0, pM1b (CPLN) | 1000 |
| KEM-025 | 0 | IVB | pT3c, pN1 (1/31), pL1, pV0, pM1b (CPLN) | 2000 |
| KEM-026* | 0 | IVB | pT3c, pN1 (7/51), pL1, pV0, pM1b (CPLN) Uterine carcinoma pT1a, G3, pL0, pV0 | 4000 |
| KEM-031 | 0 | IVB | pT3c, pN1 (4/77), pL1, pV1, pM1b (LSK) | 20 |
| KEM-001 | 1 | IVB | pT3c, pNX, pL1, pV0, pM1 (Sister Mary Joseph) | 4000 |
| KEM-006 | 1 | IVB | pT3b, pNx, pL0, pV0, pM1 (LSK) | no |
| KEM-014 | 1 | IVB | pT3c, pN1b (1/1), pL1, pV0, pM1b (spleen) | 3000 |
| KEM-018 | 1 | IVB | pT3c, pNx, pM1 (CPLN) | 50 |
| KEM-019 | 1 | IVB | pT3c, pN1b (4/8), pL1, pV0, pPn0, pM1b (HEP;Spleen) | no |
| KEM-020 | 1 | IVB | pT3c, pN0 (0/3), pL1 pV0, pM1b (LSK/Pleura) | 1000 |
| KEM-022 | 1 | IVA | pT3c, pN0 (0/2), pL1, pV0, pM1a (Pleura) | 6000 |
| KEM-027 | 1 | IVB | pT3c, pN1b (5/5), pL1, pV0, pM1b (breast) | 3500 |
| KEM-028 | 1 | IIIC | pT3c, pN1 (35/40), pL1, pV0 | 2500 |
| KEM-029 | 1 | IVB | pT3c, pNx, pL1, pV0, pM1 (LSK) | no |

RT: Residual disease

* simultanous endometrial carcinoma; all samples showed positive WT-1 IHC staining; FIGO: Fédération Internationale de Gynécologie et d' Obstétrique; TR = tumor residuals after surgery; CPLN: Cardio-phrenic lymph nodes; LSK: Abdominal wall metastases due to laparoscopy.

## Comparison of tissue and plasma genotypes and selection of a surrogate for tumor burden

Sequencing was carried out successfully in all 22 samples (20 ovarian carcinoma + 2 endometrial carcinomas of patients with synchronous ovarian and endometrial carcinoma) with the panel for FFPE and in all 99 plasma samples for the panel for ct-DNA analyses. After excluding SNPs, copy-number variations, translocations and germline variants, we found fifty-four unique non-synonymous somatic mutations in 25 genes. Thirty-eight of these being shared between FFPE and at least 1 corresponding ct-DNA sample. Sixteen and seven mutations were private to either tissue- or ct-DNA, respectively (S2 Table). From the group of 47 mutations detected in FFPE, only 23 and 25 were present in the ct-DNA at baseline and at d1, respectively. Nineteen mutations remained detectable at d4 among the 19 available samples and 12 among the available 16 samples at d10. The evolution of the perioperative MAFs of all mutations identified up to d10 were plotted for each individual patient (Fig 2) to understand whether somatic mutations could serve as a meaningful marker of tumour burden. These graphs show that the MAFs of mutations that were shared between tissue and plasma samples demonstrated homodirectional MAF changes in the ct-DNA. When multiple mutations were present in a single sample, changes in MAF were mostly unidirectional. Interestingly, mutations detected only in ct-DNA often displayed opposite MAF trends compared to mutations that were shared between tissue and blood samples. Overall, MAFs decreased with time from surgery, but some mutations showed a temporary increase at d1 or d4. For patients with complete resection and >1 gene mutation in ct-DNA, all mutations either had a significant decrease in the MAF from baseline to postoperative day 1, 4, 10, or had a very slight increase (KEM-031, *PIK3CA*). In comparison, patients with incomplete resection, most mutations (9 out of 19) had an increase in the MAF from baseline to postoperative day 1, 4, 10. Few mutations (7 out of 21) had a consistent decrease in the MAF from baseline to following days. Out of 13 patients having at least 2 mutations in separate genes, 8 patients showed a trend that was similar to the mutation in *TP53*, the other 4 genes showed a different trend: KEM-016 only had one detectable mutation in plasma at postoperative day 1 (*TP53*) the other mutation was only detected in tissue.

Mutations in *TP53* were the most frequently found in tissue (20/20 samples) and at baseline evaluation of ct-DNA (12/20 samples). The second and third most frequent mutations in ct-DNA at baseline were mutations *ERB-B2* (5/20 samples) and TSC2 mutations (4/20). Therefor it was decided to use *TP53* MAF as candidate for evaluation as surrogate for tumor burden. All 21 patient tissue samples (including 1 patient with different *TP53* mutations in an ovarian and an endometrial tumor), carried one *TP53* mutation each, corresponding to 19 unique sequencing variants. Two additional mutations were present only in plasma samples but were not found in the corresponding FFPEs. Six *TP53* mutations from 5 cases (N = 4 TR0 and N = 1 TR>0) remained undetected in the plasma at baseline but all but one re-appeared at d1. Information about *TP53* mutations including genotypes, samples and presence in consecutive samples are presented in detail in Fig 3.

## Quantification of tumour burden based on plasma markers

**Tumor genomic markers in ct-DNA.** The mean MAF of *TP53* mutations in all patients with complete resection (TR0) decreased significantly, whereas this could not be seen for patients with residual disease (TR>0). At baseline (D0), TP53 MAF did not differ among TR0 and TR>0 with mean MAF = 1.91 versus 1.73, (p = 0.86). TP53 MAFs were significantly lower for patients with TR0 compared to patients with TR>0 at post-operative day 1 (mean MAF = 0.06 versus 2.06; p = 0.002), post-operative day 4 (mean MAF = 0.07 versus 1.8; p = 0.04)

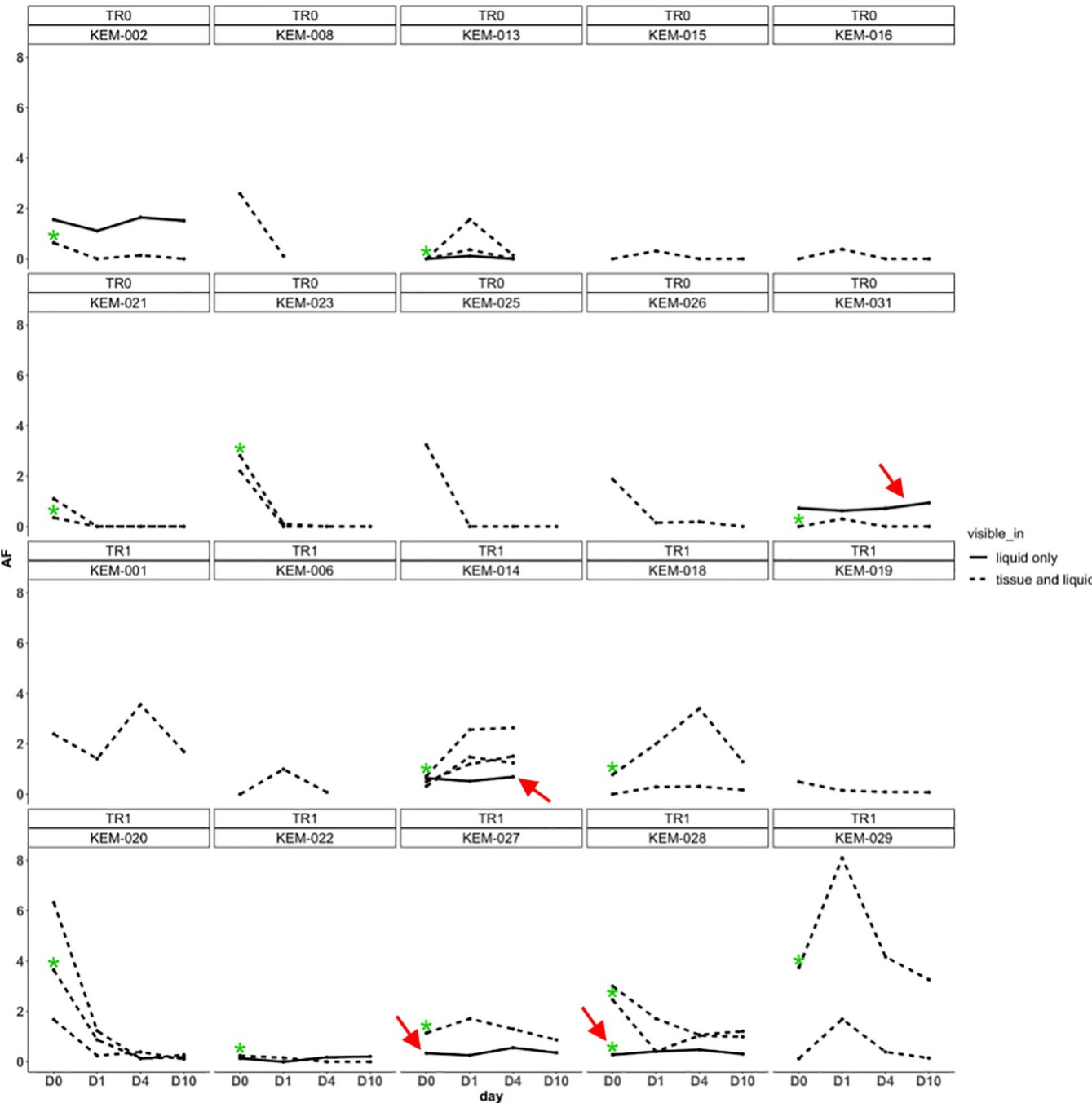

**Fig 2. MAF variation in FFPE samples and consecutive blood samples.** In the upper sections of the headings size of residual disease is displayed (TR0-macroscopic complete resection), in the row below the patient identifiers of consecutive patients with TR0 (upper both sections) and TR1 (lower both sections); Green asterisks indicate mutations in TP53. Red arrows indicate samples with different directions of MAF trends of mutations detected only in cfDNA compared to mutations that were shared between tissue and blood samples.

and post-operative day 10 (mean MAF = 0 versus 1.04; p = 0.008) (Fig 4). The difference was more pronounced at d10 and at this point of time, all assessable TR0 cases showed no evidence of *TP53* mutations. In comparison, in 7 out of 8 TR>0 cases *TP53* MAFs ranging from just below the level of detection (0.1%) up to 3.26% (Fig 4; p = 0.008) were observed at d10.

| Case | TR status | Mutation (gene protein) | | FFPE | day 0 1 4 10 | |
|------|-----------|-------------------------|--|------|--------------|--|
| 16 | 0 | ERBB2 | p.S310F | yes | | - |
| 16 | 0 | PIK3CA | p.E542A | yes | | - |
| 16 | 0 | TP53 | p.D228fs | yes | | - |
| 16 | 0 | TP53 | p.E343fs | yes | | ↓ |
| 21 | 0 | TP53 | p.R248Q | yes | | ↓ |
| 21 | 0 | ERBB2 | p.P1228L | yes | | ↓ |
| 23 | 0 | RB1 | p.Y453* | yes | | ↓ |
| 23 | 0 | TP53 | p.Y234C | yes | | ↓ |
| 15 | 0 | TP53 | p.L330fs | yes | | ↓ |
| 25 | 0 | TP53 | p.R181P | yes | | ↓ |
| 26 | 0 | TP53 | splice | yes | | ↓ |
| 31 | 0 | TSC1 | p.D802fs | yes | | ↓ |
| 31 | 0 | TP53 | p.M246fs | yes | | ↓ |
| 31 | 0 | ERBB2 | p.R816C | no | | ↑ |
| 2 | 0 | TP53 | p.R110P | yes | | ↓ |
| 2 | 0 | RB1 | p.P298fs | no | | ↓ |
| 22 | 1 | TP53 | p.R196* | yes | | ↓ |
| 22 | 1 | TP53 | p.R248Q | no | | ↑ |
| 18 | 1 | TSC2 | p.E547fs | yes | | ↑ |
| 18 | 1 | TP53 | p.R273C | yes | | ↑ |
| 28 | 1 | TP53 | p.N239D | yes | | ↑ |
| 28 | 1 | PIK3CA | p.L113_N114insIL | yes | | ↓ |
| 28 | 1 | TP53 | p.V122fs | yes | | ↓ |
| 1 | 1 | TP53 | p.V272fs | yes | | ↓ |
| 19 | 1 | TP53 | p.G244C | yes | | ↓ |
| 20 | 1 | TP53 | p.R273C | yes | | ↓ |
| 20 | 1 | TSC1 | p.Q778* | yes | | ↓ |
| 20 | 1 | BRCA2 | p.W2586* | yes | | ↓ |
| 27 | 1 | TP53 | p.G245D | yes | | ↓ |
| 27 | 1 | ERBB2 | splice | no | | ↑ |
| 29 | 1 | TSC2 | splice | yes | | ↑ |
| 29 | 1 | TP53 | p.R158G | yes | | ↓ |

**Fig 3. Correlations of mutations in FFPE and ct-DNA.** Only cases with full data are shown. Green cells denote absence and brown cells presence of mutation at a given time point. Arrows indicate differences in MAF compared to d0 or d1 when d0 measurement was not available. There were 3 cases with opposite MAF trends at d10 shown in bold. All 3 involved a mutation that were not present in the sampled FFPE.

Of note, 2 patients undergoing TR0 with undetectable *TP53* mutations at d10, showed evidence of other mutations at this timepoint. Interestingly, neither variant was present in the corresponding FFPE sample (Fig 4). Patients undergoing complete resection received non-significantly higher numbers of packed blood cells compared to patients undergoing incomplete

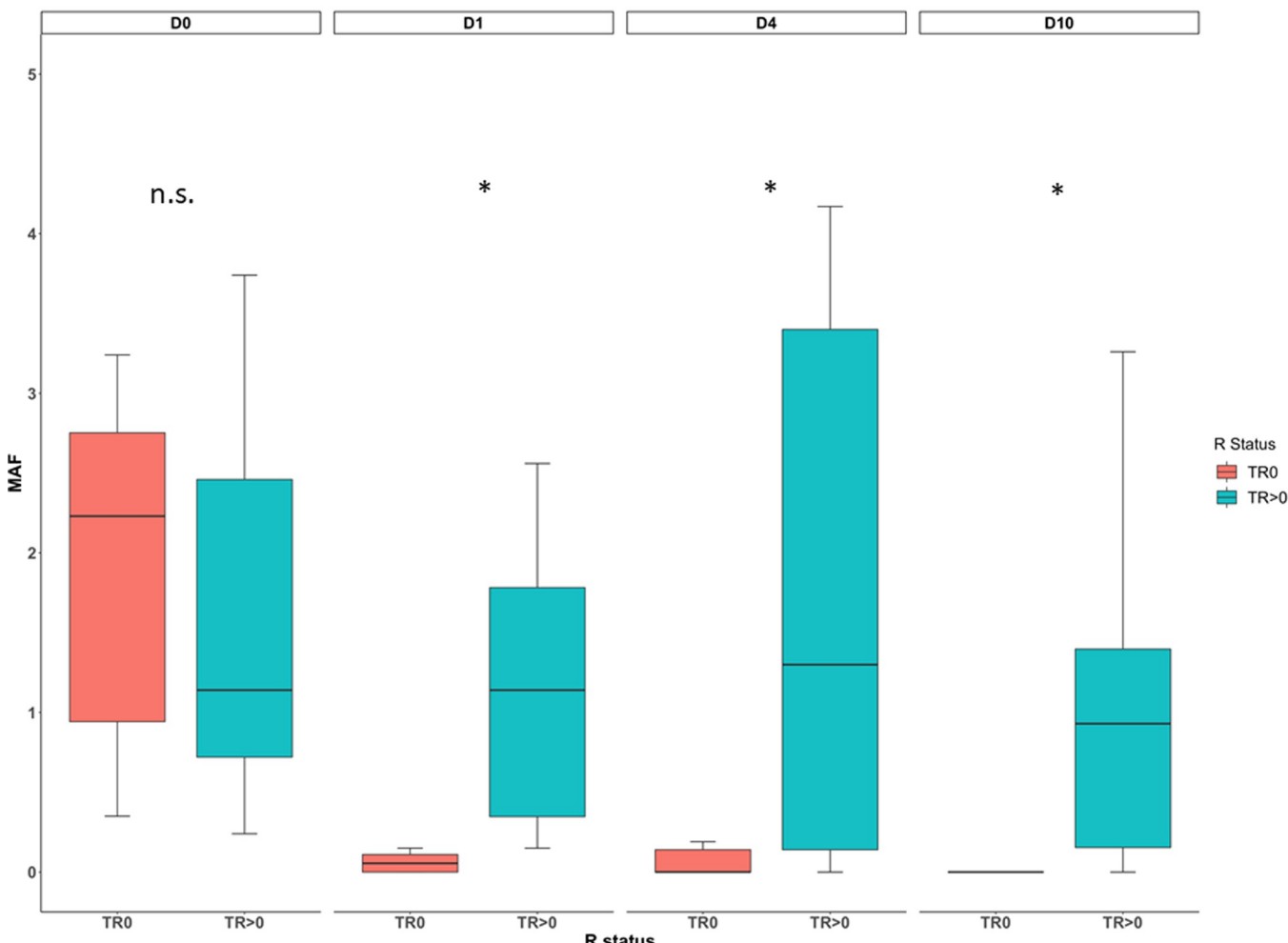

**Fig 4. *TP53* MAF of ct-DNA in dependency of debulking surgery result.** Orange is bar TR0, blue bar is TR>0; n.s. not statistically different between TR0 and TR>0; *statistically different between TR0 and TR>0.

resection (S3A Fig). However, it should be ruled out, that this was a reason for lower MAF in patients with complete resection. The Pearson's correlation coefficient was not significant for the percentage change in Δ MAF when compared to the amount of packed blood cells given between baseline and post-operative day 1, 4 or 10, respectively (S3B Fig).

**Tumor marker CA125 and HE4 in serum and ascites.** In all patients included to this study, elevated CA125 and HE4 was detected in baseline serum and ascites samples. There were no differences between baseline levels in CA125 and HE4, neither in serum, nor in ascites in patients with TR0 or TR>0. CA125 and HE4 levels in serum and ascites were significant different between patients with TR0 and TR>0 at all consecutive points of time (S1A–S1D Fig). Moreover, the differences (Δ) of logCA125 serum levels were significantly different between baseline and post-operative day 4 in patients with TR0 (p = 0.036), but not in patients with TR>0. However, ΔlogCA125 serum level between baseline and post-operative day 10 was significantly different in patients with TR>0 (p = 0.0088), but not in patients with TR0 (S1A Fig). The Δ of logHE4 serum levels were neither different between baseline and post-operative day 4 and baseline and day 10 in patients with TR0, nor in patients with TR>0. (S1B Fig). S1C Fig displays the Δ log of CA125 levels in ascites between baseline and post-operative day 4 and

post-operative day 10. In patients with TR0 Δ logCA125 levels were significant lower compared to baseline (p< 0.001) at day 4 and day 10 (p<0.01). Same was true for Δ logHE4 ascites levels (baseline to day 4; P<0.001 and baseline to day 10; p<0.001) (S1D Fig). However, in the individual absolute change of serum CA125 and HE4 (S2A and S2B Fig), nearly all patients, even with TR>0, experienced some decline after surgery. Individual absolute changes of ascites CA125 and HE4 levels indicated a homogenous and distinct decline in patients with TR0 compared to patients with TR>0 (S2C and S2D Fig).

## Discussion

In this prospective study it was shown, that MAF of *TP53* mutation detected in ct-DNA was capable of monitoring disease burden in patients with primary high-grade serous ovarian cancer (EOC). This was proven by the observation, that in all patients with EOC undergoing complete resection, complete depletion of *TP53* mutations was observed post-surgically, in comparison to complete depletion of *TP53* mutations in only one patient with residual disease. In comparison, using both serum and ascites CA125 and HE4 tumor burden could not sufficiently differentiate between patients with and without residual disease, despite showing a clear response to the surgical treatment. The lack of power to differentiate between patients with complete resection and patients with incomplete resection was mainly due to the fact, that tumor markers also decreased in patients undergoing incomplete resection. The main reason for the decrease of the established tumor markers might be the reduction of ascites in all patient, as ascites has been described as one of the most influencing factors of CA125 [21]. Moreover, the rather long half-life of CA125 (~168 hours [22])—especially in serum- reduces the ability of those markers to determine tumor burden in an effective manner. In comparison, half-life of ct-DNA in plasma is thought to be up to ~ 2h [23], making ct-DNA based markers a dynamic option in timely critical evaluation settings. CA125 and HE4 assessment before onset and shortly after onset of chemotherapy have been shown to be predictors of response to therapy and even to survival [24–26]. However, no benefit of early chemotherapy initiation, solely based on CA125 increase [6], no CA125 elevation in nearly 10% of patients with EOC [27], and a weak correlation between CA125 kinetics with tumor size determined by computer tomography [28] restrained implementation of tumor marker kinetics to guide individual therapy so far. In addition, as CA125 and HE4 are physiologically present in serum even in healthy persons, cut-off determination to differentiate between "*no active cancer*" and "*still active cancer*", or "*significant reduction in activity of cancer*" and "*non-significant reduction in activity of cancer*" in patients undergoing cancer treatment is challenging. Assessment of ct-DNA has the potential to overcome these limitations by analyzing genes, which are cancer associated, exclusively, or by analyzing genomic instability as rather global marker [29]. In here we could demonstrate, that 7 of 41 mutations found in 11 genes were detectable in ct-DNA but not FFPE tissue. This highlights the potential of ct-DNA analyses to map tumor heterogeneity better than tissue-based analyses [30]. If multi-gene NGS technology for ct-DNA analyses was used in patients with advanced EOC, further prognostic, or even predictive information from mutations (e.g. in *BRCA* genes and other genes from the *homologous recombination deficiency (HRD)* pathway) might be found to guide treatment, based on the current mutational status of a tumor. The lower detection threshold of the ct-DNA assay used in the present study was reached in all patients with macroscopic complete resection at day 10 after surgery, highlighting one of the limitations of the present study. The detection threshold of the ct-DNA assay was low enough to identify all the patients without visible disease correctly. However, it is generally known in the treatment of patients with advanced EOC, that 1st-line combination therapy and maintenance therapy is of high importance after surgery- even in

patients with complete macroscopic resection- to eradicate non-visible tumor cells. Thus, the threshold of the current ct-DNA assay was obviously not low enough to detect minimal (non-visible) disease- defined by the surgeon at the end of surgery. Consequently, further optimization of the ct-DNA assay is of interest to provide deeper insight into non-visible, minimal residual- but tumor-biologic active- disease. A retrospective study of 51 patients with recurrent ovarian cancer showed, that pre-treatment *TP53* mutational levels in ct-DNA and a decrease of the *TP53* MAF >60% between baseline and the second cycle of chemotherapy was associated with increased time to progression [18]. Thus, in patients undergoing chemotherapy without sufficient reduction in *TP53* MAF, chemotherapy might be stopped early, or specific trials could be set up to evaluate new treatment strategies in such patients. Ct-DNA assay cannot be used for the reliable detection of insertions and deletions (InDels) with a size of ~25 – 600bp. Due to misalignment events, that might occur with this assay, the allele frequency for InDels might be biased and functionally annotated synonymous mutations might result in cryptic splice sites-which is another limitation of the current study. Nevertheless, to our knowledge it is the first prospective study giving rise to an analytically valid test to objectively determine accuracy of ct-DNA as measure of tumor burden in patients with primary EOC. Therefore, this study might be the first step demanded by a recent joint review from American Society of Clinical Oncology and College of American Pathologists stating that there is insufficient evidence of clinical validity and utility for the majority of ct-DNA assays in advanced and in early-stage cancer, for treatment monitoring, or residual disease detection [31]. Furthermore, resection status of debulking surgery is an important prognostic factor for patients with advanced EOC [32] and it is one of the main stratification criteria used in clinical trials. Introduction of an objective and more valid method to determine the surgical resection status would overcome known shortcomings of surgeon determined resection status at the end of surgery [33] and might lead to better patients selection and better understanding of clinical trial results.

## Supporting information

**S1 Fig.** A: **Serum CA125 in dependency of tumor burden** (debulking status after surgery); R status- residual disease after surgery; D0 baseline; D4 day 4 after surgery; D10 day 10 after surgery; p-values report comparison between TR0 and TR>0; n.s.- not significant B: **Serum HE4 in dependency of tumor burden** (debulking status after surgery); R status- residual disease after surgery; D0 baseline; D4 day 4 after surgery; D10 day 10 after surgery; p-values report comparison between TR0 and TR>0; n.s.- not significantC: **Ascites CA125 in dependency of tumor burden** (debulking status after surgery); R status- residual disease after surgery; D0-baseline; D4 -day 4 after surgery; D10- day 10 after surgery; p-values report comparison between TR0 and TR>0; n.s.- not significantD: **Ascites HE4 in dependency of tumor burden** (debulking status after surgery); R status- residual disease after surgery; D0 baseline; D4 day 4 after surgery; D10 day 10 after surgery; p-values report comparison between TR0 and TR>0; n.s.- not significant.
(DOCX)

**S2 Fig. Individual absolute change of serum and ascites CA125 and HE4 across all patients with complete (TR0, orange)) and incomplete resection (TR>0, blue) between baseline and day 4(d4) and day10 (d10).** A CA125 in Serum; B HE4 in Serum; C CA125 in ascites; D HE4 in ascites.
(DOCX)

**S3 Fig.** a **Number of packed blood** units given between atients with TR0 and TR>0. b: **Correlation between packed blood units and TP53** mutations depending TR0 and TR>0 at the post-surgical days.
(DOCX)

**S1 Table.  a: Genes included on the panel for ct-DNA analyses**; b: **Amplicons targeted** for sequencing of the genes included on the panel for ct-DNA analyses.
(DOCX)

**S2 Table. Mutations found in tumor genome and at least one corresponding ct-DNA samples of each patient; yellow-coloured lines indicate private tissue- and blue-cloured lines indicate private ct-DNA mutations.**
(DOCX)

## Author Contributions

**Conceptualization:** Florian Heitz, Sotirios Lakis, Roopika Menon, Judith N. Müller, Alexandra Hengsbach, Lukas C. Heukamp, Andreas du Bois, Johannes M. Heuckmann.

**Data curation:** Florian Heitz, Sotirios Lakis, Sebastian Heikaus, Roopika Menon, Beyhan Ataseven, Stephanie Schneider, Erika Mariotti, Mareike Bommert, Sonia Prader, Frauke Leenders, Alexandra Hengsbach, Christian Gloeckner, Andreas du Bois.

**Formal analysis:** Florian Heitz, Sotirios Lakis, Jalid Sehouli, Jatin Talwar, Miriam Bertrand, Judith N. Müller, Elena Ioana Braicu, Lukas C. Heukamp, Johannes M. Heuckmann.

**Funding acquisition:** Florian Heitz, Sotirios Lakis, Lukas C. Heukamp, Johannes M. Heuckmann.

**Investigation:** Florian Heitz, Sotirios Lakis, Philipp Harter, Jalid Sehouli, Jatin Talwar, Miriam Bertrand, Erika Mariotti, Elena Ioana Braicu, Andreas du Bois, Johannes M. Heuckmann.

**Methodology:** Florian Heitz, Sotirios Lakis, Roopika Menon, Miriam Bertrand, Alexandra Hengsbach, Elena Ioana Braicu, Lukas C. Heukamp, Johannes M. Heuckmann.

**Project administration:** Florian Heitz, Alexandra Hengsbach.

**Software:** Jatin Talwar.

**Supervision:** Florian Heitz.

**Visualization:** Florian Heitz.

**Writing – original draft:** Florian Heitz, Sotirios Lakis, Sebastian Heikaus, Jalid Sehouli, Jatin Talwar, Roopika Menon, Beyhan Ataseven, Stephanie Schneider, Erika Mariotti, Mareike Bommert, Judith N. Müller, Sonia Prader, Frauke Leenders, Alexandra Hengsbach, Christian Gloeckner, Elena Ioana Braicu, Lukas C. Heukamp, Andreas du Bois, Johannes M. Heuckmann.

**Writing – review & editing:** Florian Heitz, Sotirios Lakis, Philipp Harter, Sebastian Heikaus, Jalid Sehouli, Jatin Talwar, Roopika Menon, Beyhan Ataseven, Miriam Bertrand, Stephanie Schneider, Erika Mariotti, Mareike Bommert, Judith N. Müller, Sonia Prader, Frauke Leenders, Alexandra Hengsbach, Christian Gloeckner, Elena Ioana Braicu, Lukas C. Heukamp, Andreas du Bois, Johannes M. Heuckmann.

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
