## [Decision Letter · Decision Letter 0]

7 Jul 2021

PONE-D-21-13703

Cell-free tumor DNA, CA125 and HE4 for the objective assessment of tumor burden in patients with advanced high-grade serous ovarian cancer.

PLOS ONE

Dear Dr. Heitz,

Thank you for submitting your manuscript to PLOS ONE. After careful consideration, we feel that it has merit but does not fully meet PLOS ONE’s publication criteria as it currently stands. Therefore, we invite you to submit a revised version of the manuscript that addresses the points raised during the review process.

Specifically, the method section will need a more detailed description of the experimental procedures used and the result section will need to be rewritten to improve clarity.

We look forward to receiving your revised manuscript.

Kind regards,

Kwong-Kwok Wong, Ph.D.

Academic Editor

PLOS ONE

Journal Requirements:

This study was supported by NewOncology, Cologne, Germany

This study was supported by NewOncology, Cologne, Germany. The funders played a role in study design, and analysis, and preparation of the manuscript.

FH: Travel grants: AstraZeneca, Tesaro, Roche; Honoraria: Roche, AstraZeneca; Clovis, Advisory: Roche; SL: personal fees from NEO New Oncology GmbH, personal fees from BioNTech Diagnostics, personal fees from Definiens GmbH; PH: Honoraria: Roche, AstraZeneca, Tesaro; Advisory: Roche, AstraZeneca, Tesaro, PharmaMar, Lilly; SH: none; 

JS: HONORARIA: Astra Zeneca, Eisai, Clovis, Olympus, Johnsons and Johnson, PharmaMar, Pfizer, TEVA, TESARO, MSD; CONSULTING OR ADVISORY ROLE: Astra Zeneca, Clovis, Lilly, PharmaMar, Pfizer, Roche, TESARO, MSD; RESEARCH FUNDING: Astra Zeneca, Clovis, Merck, Bayer, PharmaMar, Pfizer, TESARO, MSD; TRAVEL, ACCOMODATIONS, EXPENSES: Astra Zeneca, Clovis, PharmaMar, Roche, Pfizer, TESARO, MSD; JT: employed at New Oncology; RM: employed at New Oncology; BA: reports receiving honoraria from Roche, Tesaro, Clovis, AstraZeneca, and Celgene for lectures, and is an unpaid consultant/advisory board member for Roche and Amgen; MB employed at New Oncology; 

SS: none; EM: employed at New Oncology; MB: Travel support from prIME Oncology; JNM: employed at New Oncology; SP: none; FL: employed at New Oncology; AH: none; CG: employed at New Oncology; EIB: reports receiving honoraria for advisory board and educational activities from AstraZeneca, Clovis, Tesaro, GSK, Roche Pharma, Incyte, Eisai, MSD, Abbvie; reports receiving travel costs from Clovis, Tesaro, Roche Pharma; LCH: employed at New Oncology; AdB: reports honorary for advisory board and educational activities for Roche, Astra Zeneca, Tesaro, Clovis, Biocad, and Genmab; JMH: employed at New Oncology

Reviewers' comments:

Reviewer's Responses to Questions

**Comments to the Author**

1. Is the manuscript technically sound, and do the data support the conclusions?

Reviewer #1: Yes

Reviewer #2: Partly

2. Has the statistical analysis been performed appropriately and rigorously? 

Reviewer #1: I Don't Know

Reviewer #2: No

3. Have the authors made all data underlying the findings in their manuscript fully available?

Reviewer #1: Yes

Reviewer #2: No

4. Is the manuscript presented in an intelligible fashion and written in standard English?

Reviewer #1: Yes

Reviewer #2: No

5. Review Comments to the Author

Reviewer #1: The authors present a novel approach for liquid biopsy-based monitoring of HGSC progression.

The idea is innovative and the application setting is pertinent. Nonetheless I have some comments that if addressed might improve the quality of this manuscript.

1)When relating on the coverage of the sequencing please mention the fragment sizes of the ct DNA (which is usually very fragmented).

2)Was the panel of genes a custom panel? If so, how were the genes selected?

3) is the filtering out of variants with MAF between 50-100% a common practice? Please provide references for this approach

4)Does R>0 refers to macroscopic disease residual? (> 1 cm?)

5)I suggest to provide FIGO stage in table

6)Would be interesting to address the reason for having mutations in ctDNA non present in FFPE DNA in discussion

7)Figure 2 legend (and in general all legends) is too long,maybe it should be shortened and additional text should go in results or discussion

8)is MAF reflecting a low ctDNA yield in general?

9)Is any patients doing chemo during the time of blood sampling?

10)Are blood samples 20 ml each? seems a lot...is it a typo?

11)Fig 3 legend: green cells are described twice and orange cells are not

In general the results section seems a bit confused. The first paragraph does not have a title. Second paragraph's title ends with "selection of a surrogate for tumor burden" ? What does it mean? Is TP53 allele frequency identified as the tumor burden marker?What about the other genes?

"Quantification of tumor burden based on liquid markers"= tumor burden is not quantified in this study, this is an extreme assumption. The only possible thing is inferring on tumor residual presence.

Last paragraph of results says "Tumor marker CA125/HE4.." as if the considered marker is the ratio between these two but instead the two of them are considered separately.

Paragraphs and sub-paragraphs arrangement in Results is not flowing smoothly

In Methods statistics is not described at all and it should be.

Reviewer #2: The study by Heitz et al. reports on a prospective study monitoring of ctDNA in HGSOC undergoing surgical resection and contrasted with serum CA125 and HE4. The results are very encouraging as they suggest that ctDNA correlates with residual disease based on complete or incomplete resection.

However, the way the data was presented is confusing. This precludes a clear comparison of the performance of ctDNA relative to common markers such as CA125 and HE4. Overall, the manuscript requires extensive editing to facilitate understanding of the work performed, plus some grammatical revisions.

Specific comments

1. It is not so clear whether the patients were newly diagnosed/treatment naïve before commencement of the debulking surgery. Also, during the post-operative surveillance, were the patients undergoing adjuvant chemo? Please provide details.

2. At lines 142 and 146; the extraction process, the cfDNA has not yet been analysed as mutant DNA, hence technically, should be acknowledged as cfDNA, rather than ctDNA. Furthermore, did you do any quantification of the extracted cfDNA? If no, explain why that was not done, as their levels at different time-points could potentially provide an important information of total cfDNA changes at pre- and post- surgery.

3. At line 139, provide details of the DNA extraction protocol for the FFPE?

4. At line 141, specify how much plasma was used for cfDNA extraction.

5. Details of the amplicons targeted for sequencing in each one of the genes should eb provided beyond the list in Table S1 (Table S1 labelled S3 Table in this submission).

6. Further details in the method need to be provided: Does the panel uses Unique molecular identifier? Detail on the manufacturer’s providers of library preparation kit and hybrid capture oligos.

7. In the result section and other parts of the manuscript, it would be more appropriate to specify the word ‘plasma’, not ‘liquid’, when comparing the ctDNA mutations with tissue-derived data.

8. In Figure 1, why do you refer to serum availability if ctDNA was measured in plasma. In any case should be noted both.

9. The figure legends should commence with a figure title.

10. The legend of figure 1 is over extensive but fail to provide sufficient information to understand the figure. Please describe the symbols used in the figure. For example in Figure 1: what do the red arrows exemplify?, in what order are the cases displayed. Do not provide any discussion of the results on the legend. That should be in the text.

11. In Supplement 4 Table ( refer in text as S2 Table – I believe), please indicate what ‘ressource’ means as a column heading? cf DNA – correct to nucleotide change. All genes should be in italics.

12. It is unclear from S4 (S2) Table to determine if mutation were found in cfDNA at baseline or day 1 or both, as indicated in the text; or if detected later on.

13. No description of the mutations found in tissue only or blood only is given, or not very clear if presented.

14. To ascertain the drop in ctDNA that the authors refer to in Figure 3, either the ctDNA concentrations should be provided or the colour should be toned accordingly. Otherwise is not apparent that ctDNA decrease in a number of TR1 cases.

15. In figure 4 – indicate the statistical comparisons in the graph. Again, details should be provided in the text not in figure legend. Please indicate what statistical test was performed – whether parametric/non-parametric, paired?, two-sided?

16. The statement in lines 240-242 is better referred to Figure 4 for demonstration, than in Figure 3.

17. Please add the data associated with statements in lines 243-248 as supplementary.

18. The statistics carried out to compare the DeltalogCA125 and HE4 are not clear at all. Neither the data in Figure S1. Could you please represent the data grouped in the same manner that the ctDNA in figure 4?

19. In Figure S1 – there is no asterisks in the ascites CA125 and HE4 comparison, even though the text it says there are significant statistical differences.

6. PLOS authors have the option to publish the peer review history of their article (what does this mean?). If published, this will include your full peer review and any attached files.

Reviewer #1: No

Reviewer #2: **Yes: **Elin Gray

---

## [Author Response · Author response to Decision Letter 0]

10 Oct 2021

Reviewers’ comments and responses

We included to responses to reviewers suggestions below. The lines denoted are based on the red-version of the R1-version of the manuscript. 

Reviewer #1: The authors present a novel approach for liquid biopsy-based monitoring of HGSC progression.

The idea is innovative and the application setting is pertinent. Nonetheless I have some comments that if addressed might improve the quality of this manuscript.

1)When relating on the coverage of the sequencing please mention the fragment sizes of the ct DNA (which is usually very fragmented). Added to ll.166-175 (in accordance with comment 2 from reviewer 2): After extraction the fragment size for each sample was determined by a distinct peak which was obtained by running the samples through a Fragment Analyser (Agilent). The instrument uses a capillary electrophoresis-based separation technique and provides peaks based on the fragment sizes present in the sample. The fragment size of the cfDNA ranged between 160-180bp based on the nucleosome cleavage sites. The amount of cfDNA in the sample includes both DNA that is shed from the tumor (ctDNA) and normal cells. A clear indication of the tuor DNA in the sample can be judged by the MAFs. However, recent publications have also shown the utility of simple cfDNA concentration measurement (20), thus mutation allele frequency is a more precise measure of residual tumor compared to cfDNA and offers the advantage of genotyping. 

2)Was the panel of genes a custom panel? If so, how were the genes selected? May you provide this? added to ll. 185-189: “The panel of genes for the NEOliquid test were part of a CE kit produced by NEO New Oncology GmbH.” please find further information in the response to number 3 below.

3) is the filtering out of variants with MAF between 50-100% a common practice? Please provide references for this approach � ll 185-189 rephrased and added: ”The NEOliquid test was designed to identify genes either with a direct or indirect impact on patient treatment decisions for solid tumors. Therefore, core bioinformatics analysis was set up to exclude variants with MAF in the range of 50% or 100% in liquid samples which were highly unlikely to be somatic in origin. Manual curation was needed only in select cases and were therefore excluded.”

4)Does R>0 refers to macroscopic disease residual? (> 1 cm?)- no TR0 means macroscopic complete resection as described in the 2nd sentence of the results section: ”Ten patients with macroscopic complete resection (TR0) and ten patients with residual disease (TR>0)…”- but now additionally added to ll.- 132-136:” For the purpose of this proof-of-principle study the determination of “tumor burden” was based upon the post-surgical result of debulking surgery. Patients without macroscopic residual disease were defined as TR0 (“low tumor burden”), patients with any macroscopic residual disease were defined as TR>0 (“high tumor burden”).”

5)I suggest to provide FIGO stage in table- it is not clear in which table FIGO should be included? It was already included to table 1, see column 3. 

6)Would be interesting to address the reason for having mutations in ctDNA non present in FFPE DNA in discussion� added to the discussion ll 373-376: ”In here we could demonstrate, that 7 of 41 mutations found in 11 genes were detectable in ct-DNA but not FFPE tissue. This highlights the potential of ct-DNA analyses to map tumor heterogeneity better than tissue-based analyses (30).”

7)Figure 2 legend (and in general all legends) is too long, maybe it should be shortened and additional text should go in results or discussion� shortened and transferred to the results section, ll. 232-245:” When multiple mutations were present in a single sample, changes in MAF were mostly unidirectional. Interestingly, mutations detected only in cfDNA often displayed opposite MAF trends compared to mutations that were shared between tissue and blood samples. Overall, MAFs decreased with time from surgery, but some mutations showed a temporary increase at d1 or d4. For patients with complete resection and >1 gene mutation in cfDNA, all mutations either had a significant decrease in the MAF from baseline to postoperative day 1, 4, 10, or had a very slight increase (KEM-031, PIK3CA). In comparison, patients with incomplete resection, most mutations (9 out of 19) had an increase in the MAF from baseline to postoperative day 1, 4, 10. Few mutations (7 out of 21) had a consistent decrease in the MAF from baseline to following days. Out of 13 patients having at least 2 mutations in separate genes, 8 patients showed a trend that was similar to the mutation in TP53, the other 4 genes showed a different trend: KEM-016 only had one detectable mutation in liquid at postoperative day 1 (TP53) the other mutation was only detected in tissue.”

8)is MAF reflecting a low ctDNA yield in general? 

In liquid biopsies it would be difficult to know exactly what fraction of the cfDNA is coming from tumor or non tumor DNA fragments, the low MAF might be indicative of low amount of ctDNA in the sample. 

9)Is any patients doing chemo during the time of blood sampling? � no. included to the method section ll. 143-144: “No cancer-specific systemic therapy was given to the patients during surgery or in the post-surgical days during the study.””

10)Are blood samples 20 ml each? seems a lot...is it a typo?- the correct amount of blood samples taken were 18ml. 

11)Fig 3 legend: green cells are described twice and orange cells are not- changed: ”Green cells denote absence and brown cells presence of a mutation at a given time point.”

In general the results section seems a bit confused. The first paragraph does not have a title.  added: ”Patients’ characteristics” 

Second paragraph's title ends with "selection of a surrogate for tumor burden" ? What does it mean? Is TP53 allele frequency identified as the tumor burden marker? What about the other genes? added to ll 275-278: ”Mutations in TP53 were the most frequently found in tissue (20/20 samples) and at baseline evaluation of ct-DNA (12/20 samples). The second and third most frequent mutations in ct-DNA at baseline were mutations ERB-B2 (5/20 samples) and TSC2 mutations (4/20). Therefor it was decided to use TP53 MAF as candidate for evaluation as surrogate for tumor burden. ” 

"Quantification of tumor burden based on liquid markers"= tumor burden is not quantified in this study, this is an extreme assumption. The only possible thing is inferring on tumor residual presence.  clarified in Methods section ll. 138-144:” For the purpose of this proof-of-principle study the determination of “tumor burden” was based upon the post-surgical result of debulking surgery. Patients without macroscopic residual disease were defined as TR0 (“low tumor burden”), patients with any macroscopic residual disease were defined as TR>0 (“high tumor burden”). An analogue definition has been published earlier (20).”

Last paragraph of results says "Tumor marker CA125/HE4.." as if the considered marker is the ratio between these two but instead the two of them are considered separately.  changed to:” Tumor marker CA125 and HE4 in serum and ascites”

Paragraphs and sub-paragraphs arrangement in Results is not flowing smoothly� we have restructured the results section in parts, that we feel, that it reads very good. 

In Methods statistics is not described at all and it should be. � included to the Method section, ll. 193-200: ” The unpaired two-sided wilcoxon rank sum test was used to determine the significance of changes across multiple days for patients paired by debulking status. A non-parametric test was choosen as the data was skewed towards low numbers (because of the decrease in MAF and serum and ascites values). To analyse the CA125 and HE4 data, values were logarithmized due to non-normal distribution. The Pearson correlation coefficient was used to measure the correlation between two sets of data. Analyses were conducted using the rstatix R-package.”

Reviewer #2: The study by Heitz et al. reports on a prospective study monitoring of ctDNA in HGSOC undergoing surgical resection and contrasted with serum CA125 and HE4. The results are very encouraging as they suggest that ctDNA correlates with residual disease based on complete or incomplete resection.

However, the way the data was presented is confusing. This precludes a clear comparison of the performance of ctDNA relative to common markers such as CA125 and HE4. Overall, the manuscript requires extensive editing to facilitate understanding of the work performed, plus some grammatical revisions.

Specific comments

1. It is not so clear whether the patients were newly diagnosed/treatment naïve before commencement of the debulking surgery.  it is claryfied in l. 108 of the Methods-Section:” Patients with treatment-naive known or highly suspected advanced (FIGO IIIC or IV) high-grade serous EOC were scheduled for primary debulking surgery”. 

Also, during the post-operative surveillance, were the patients undergoing adjuvant chemo? Please provide details. � included to the method section ll. 143-144: No cancer-specific systemic therapy was given to the patients during surgery or in the post-surgical days during the study.””

2. At lines 142 and 146; the extraction process, the cfDNA has not yet been analysed as mutant DNA, hence technically, should be acknowledged as cfDNA, rather than ctDNA. � changed. 

Furthermore, did you do any quantification of the extracted cfDNA? If no, explain why that was not done, as their levels at different time-points could potentially provide an important information of total cfDNA changes at pre- and post- surgery. � added to the Methods section ll 166-175: “After extraction the fragment size for each sample was determined by a distinct peak which was obtained by running the samples through a Fragment Analyser (Agilent). The instrument uses a capillary electrophoresis-based separation technique and provides peaks based on the fragment sizes present in the sample. The fragment size of the cfDNA ranged between 160-180bp based on the nucleosome cleavage sites. The amount of cfDNA in the sample includes both DNA that is shed from the tumor (ctDNA) and normal cells. A clear indication of the tumor DNA in the sample can be judged by the MAFs. However, recent publications have also shown the utility of simple cfDNA concentration measurement (20), thus mutation allele frequency is a more precise measure of residual tumor compared to cfDNA and offers the advantage of genotyping.”

3. At line 139, provide details of the DNA extraction protocol for the FFPE?

Added: DNA was extracted using a semi-automated extraction protocol (Maxwell®16, Promega)

4. At line 141, specify how much plasma was used for cfDNA extraction. � added to Method section LL 158-159: “For the purpose of DNA extraction, depending on the sample, plasma from 3-15ml was obtained”

5. Details of the amplicons targeted for sequencing in each one of the genes should eb provided beyond the list in Table S1 (Table S1 labelled S3 Table in this submission� changed). Included as Table S1b

6. Further details in the method need to be provided: Does the panel uses Unique molecular identifier? � yes, included to the method section ll 161-164:” In brief, after cf-DNA extraction, adapters unique molecular identifiers were ligated and individual genomic regions of interest were enriched using complementary bait sequences (hybrid-capture procedure).” 

Detail on the manufacturer’s providers of library preparation kit and hybrid capture oligos. � more infos added to Methods section ll 166-178.

7. In the result section and other parts of the manuscript, it would be more appropriate to specify the word ‘plasma’, not ‘liquid’, when comparing the ctDNA mutations with tissue-derived data. � Thank you very much, reads much better.

8. In Figure 1, why do you refer to serum availability if ctDNA was measured in plasma. In any case should be noted both. � cf-DNA was measured in plasma, Ca125 and HE4 in serum. Therefore, both is noted.

9. The figure legends should commence with a figure title.  included

10. The legend of figure 1 is over extensive but fail to provide sufficient information to understand the figure. Rev refers probably to Figure 2. Figure legend changed accordingly, and parts included to results section� see also reviewer 1, remark 7

Please describe the symbols used in the figure. For example in Figure 1: what do the red arrows exemplify? added to the legend Figure 2:” Red arrows indicate samples with different directions of MAF trends of mutations detected only in cfDNA compared to mutations that were shared between tissue and blood samples. ”

, in what order are the cases displayed. Do not provide any discussion of the results on the legend. That should be in the text. � Figure legend changed accordingly, and parts included to results section� see also reviewer 1, remark 7

11. In Supplement 4 Table ( refer in text as S2 Table – I believe� yes changed, thanks!), please indicate what ‘ressource’ means as a column heading? � changed to “examined tissue” as three patients had synchronous ovarian and endometrial carcinoma; cf DNA – correct to nucleotide change� changed . All genes should be in italics. � changed

12. It is unclear from S4 (S2) Table to determine if mutation were found in cfDNA at baseline or day 1 or both, as indicated in the text; or if detected later on. � S2 table was revised and clarified. Legend : ” Mutations found in tumor genome and at least one corresponding ct-DNA samples of each patient; yellow coloured lines indicate private tissue- and blue lines indicate private ct-DNA mutations.”. As we reported the somatic mutations exclusively, we excluded the presented calls from SNPs, copy-number variations, and translocations. During the review process we noted an inadequate description of the mutations found, as we only reported the mutations coming from ct-DNA-analyses and comparing with tissue DNA analyses. In the updated version we describe the shared mutations, and the private mutations to ct-DNA and tissue-DNA, resulting in slightly different numbers. Therefore, the descriptive sentences in the results section were rephrased to clarify the statement, ll. 212-225: ”After excluding SNPs, copy-number variations, translocations and germline variants, we found fifty-four unique non-synonymous somatic mutations in 25 genes. Thirty-eight of these being shared between FFPE and at least 1 corresponding ct-DNA sample, whereas four sixteen and seven mutations were private to either tissue- or ct-DNA, respectively (S2 Table). From the group of 47 mutations…”

13. No description of the mutations found in tissue only or blood only is given, or not very clear if presented. � Supplement Table 2 is re-formatted and differences are better to understand

14. To ascertain the drop in ctDNA that the authors refer to in Figure 3, either the ctDNA concentrations should be provided or the colour should be toned accordingly. Otherwise is not apparent that ctDNA decrease in a number of TR1 cases. � we do not understand remark. The arrows in Figure 3 indicate the drop in all patients. Please give further details. 

15. In figure 4 – indicate the statistical comparisons in the graph. � included

Again, details should be provided in the text not in figure legend. � done, included to results section.

Please indicate what statistical test was performed – whether parametric/non-parametric, paired?, two-sided? � added to the Methods section ll. 193-200:” Statistics: The two-sided wilcoxon rank sum test was used to determine the significance of changes across multiple days for patients paired by debulking status. A non-parametric test was choosen as the data was skewed towards low numbers (because of the decrease in MAF values).”

16. The statement in lines 240-242 is better referred to Figure 4 for demonstration, than in Figure 3. � thank you for this suggestion, changed!

17. Please add the data associated with statements in lines 243-248 as supplementary.  included into S3 figure 1 and S3 figure 2.

18. The statistics carried out to compare the DeltalogCA125 and HE4 are not clear at all. added to statistics part.

 Neither the data in Figure S1. Could you please represent the data grouped in the same manner that the ctDNA in figure 4? � changed accordingly and build four new graphs which represent Δlog values of CA125 and HE4 in serum and ascites, respectively. Description of new results, regarding comparisons between TR0 and TR>0 included to results, ll.326-329 

19. In Figure S1 – there is no asterisks in the ascites CA125 and HE4 comparison, even though the text it says there are significant statistical differences.  thanks! Checked and included.

---

## [Editor Report · Decision Letter 1]

5 Jan 2022

Cell-free tumor DNA, CA125 and HE4 for the objective assessment of tumor burden in patients with advanced high-grade serous ovarian cancer.

PONE-D-21-13703R1

Dear Dr. Heitz,

We’re pleased to inform you that your manuscript has been judged scientifically suitable for publication and will be formally accepted for publication once it meets all outstanding technical requirements.

Kind regards,

Kwong-Kwok Wong, Ph.D.

Academic Editor

PLOS ONE

Additional Editor Comments (optional):

The authors have adequately addressed reviewers' comments and is now acceptable for publication.
---

## [Editor Report · Acceptance letter]

12 Jan 2022

PONE-D-21-13703R1 

Cell-free tumor DNA, CA125 and HE4 for the objective assessment of tumor burden in patients with advanced high-grade serous ovarian cancer. 

Dear Dr. Heitz:

I'm pleased to inform you that your manuscript has been deemed suitable for publication in PLOS ONE. Congratulations! Your manuscript is now with our production department. 

Kind regards, 

on behalf of

Dr. Kwong-Kwok Wong 

Academic Editor

PLOS ONE